# A Study of the Structural Characteristics of Titanium Alloy Products Manufactured Using Additive Technologies by Combining the Selective Laser Melting and Direct Metal Deposition Methods

**DOI:** 10.3390/ma12193269

**Published:** 2019-10-08

**Authors:** Marina Samodurova, Ivan Logachev, Nataliya Shaburova, Olga Samoilova, Liudmila Radionova, Ramil’ Zakirov, Kirill Pashkeev, Vyacheslav Myasoedov, Evgeny Trofimov

**Affiliations:** 1Department of Metal Forming, South Ural State University, Lenin prospect 76, Chelyabinsk 454080, Russia; samodurovamn@susu.ru (M.S.); radionovalv@susu.ru (L.R.); 2“Composite” Joint Stock Company, 4 Pioneer Street, Korolev, Moscow Region 141070, Russia; ivan@logachev.biz; 3Department of Materials Science, Physical and Chemical Properties of Materials, South Ural State University, Lenin prospect 76, Chelyabinsk 454080, Russia; samoylova_o@mail.ru (O.S.); tea7510@gmail.com (E.T.); 4Scientific and Educational Center “Engineering”, South Ural State University, Lenin prospect 76, Chelyabinsk 454080, Russia; zakirovra@susu.ru; 5Resource Center for Special Metallurgy, South Ural State University, Lenin prospect 76, Chelyabinsk 454080, Russia; pashkeevki@susu.ru (K.P.); vmyasoedov74@mail.ru (V.M.)

**Keywords:** additive manufacturing, titanium alloys, microstructure

## Abstract

Titanium alloy product manufacturing is traditionally considered to be a rather difficult task. Additive manufacturing technologies, which have recently become quite widespread, can ensure the manufacture of titanium alloys products of an arbitrary geometrical shape. During this study, we have developed a methodology for manufacturing titanium alloys products using additive technologies on FL-Clad-R-4 complex of laser melting of metals by combined Selective Laser Melting (SLM) and Direct Metal Deposition (DMD) methods. Ti–6Al–4V and Ti–6Al–4Mo–1V alloys were used for the manufacture of samples. We studied the microstructure of the obtained details and measured the microhardness of the samples. We discovered a gradient of the structure throughout the height of the details walls, which is connected with the peculiarities of thermal cycles of the technology used. This affected the microhardness values: in the upper part of the details, the microhardness is 10–25% higher (about 500 *HV*) than in the lower part (about 400 *HV*). Products made according to the developed technique do not have visible defects and pores. The obtained results indicate the competitiveness of the proposed methodology.

## 1. Introduction

Manufacturing products from titanium and its alloys using traditional methods is a complex process task. Thus, details should be manufactured by the casting method in a controlled atmosphere (under vacuum or in a protective gas environment) to avoid the interaction of a titanium-based melt with oxygen and air nitrogen [1,2,3,4]. Further machining of such details requires special conditions and equipment and is rather costly [5,6,7,8,9,10]. In this regard, the use of additive manufacturing technologies [11,12,13,14,15,16,17,18,19,20] seems to be a very promising line, since it makes it possible to create products, of a complex geometrical shape as well, from titanium and its alloys with subsequent relatively shallow machining.

Several studies have been devoted to the use of additive technologies in the manufacture of titanium alloy products [21,22,23,24,25,26,27]. In the studies Baufeld et al. [22,23,24] analyzed the samples of titanium alloys obtained by the Shaped Metal Deposition (SMD) method. The authors indicate that the SMD method makes it possible to manufacture solid details from Ti–6Al–4V alloy both in the form of hollow cylinders and with a square cross section with the wall thickness from 5 to 20 mm. Moreover, the resulting products showed good strength performance. In the work Dinda et al. [25] studied an implant made from a Ti–6Al–4V alloy made by the Direct Metal Deposition (DMD) method. According to the data of [25], the implant initially had the necessary morphology, but revealed the required mechanical characteristics only after additional heat treatment. Kelly and Kampe in the work [26] showed the evolution of the microstructure in the products created by the Laser Metal Deposition (LMD) method. As concluded by the authors of [26], the differences in the characteristics of the microstructure throughout the height of the product wall are explained by thermal effects during the manufacture of the parts. Kobryn and Semiatin in [27] carried out a comparative analysis of Ti–6Al–4V alloy samples manufactured by pressure casting and laser melting methods. According to the results of [27], in the case of laser melting, the properties of the obtained samples strongly depend on the magnitude of the used laser power.

Thus, in the literature there are still a lack of optimal technology for manufacturing titanium alloy products with the specified properties, which requires additional research and development of new techniques. The purpose of this work is to develop a methodology for manufacturing titanium alloy details of an arbitrary geometrical shape using additive technologies by the combined SLM and DMD methods, and further study the structure and microhardness of the obtained products.

## 2. Materials and Methods

Substrates of a simple geometric shape were maded by the SLM method at the “Composite” Joint Stock Company (Korolev, Russia). However, the small size of the working camera of the equipment available at “Composite” Joint Stock Company did not allow growing on these substrates the details of the necessary geometric shape specified by the technical specifications of the sizes. Therefore, further work was continued on the basis of South Ural State University (Chelyabinsk, Russia) in order to determine the possibility of combining SLM and DMD methods for the manufacture of details from titanium alloys (including various compositions) of complex geometric shapes. 

Details were fabricated by DMD method on FL-Clad-R-4 metal laser melting complex on previously manufactured SLM substrates. The FL-Clad-R-4 complex (Figure 1) is used to work in the air medium when the inert gas is fed into the melting zone together with the transported powder. The main unit of the complex can be represented as follows: 1) 4 kW laser head with an ytterbium fiber-optics laser (LS-4); 2) KUKA R-120 six-axis robot-equipped manipulator combined with KUKA DKP-400 double-axis positioning element; 3) TWIN-10-CR-2 powder feeder with a four-axis powder feed module; 4) process chamber — a metal cylinder with the diameter of 600 mm and the length of 1100 mm. It should be noted that the initial design of the complex lacked the process chamber, and we added it to carry out experiments with titanium alloys.

All substrates fabricated by the SLM method were made of Ti–6Al–4V alloy powder. The grades of powder alloys and the mode used for melting by DMD method are given in Table 1. The chemical composition of the Ti–6Al–4V powder is: 6.2% Al, 4.5% V, 0.2% Fe, 0.25% Zr, 88.85% Ti (all in wt%). The content of elements in the Ti–6Al–4Mo–1V powder is: 5.8% Al, 1.1% V, 0.21% Fe, 0.25% Zr, 3.4% Mo, 89.24% Ti (all in wt%).

The growth height of the part wall over one pass (by DMD method) of the laser head was 0.3 mm. When manufacturing of details No. 1 and 2, the process chamber was not insulated from the outside, in other cases (details No. 3, 4 and 5) it was insulated with a heat insulator.

The microstructure of the samples were studied on Axio Observer D1.m optical inverted metallographic microscope (Carl Zeiss Microscopy GmbH, Jena, Germany) equipped with Thixomet Pro software (Thixomet Pro, Thixomet Company, Saint Petersburg, Russia) and hardware suite (Thixomet Pro, Thixomet Company, Saint Petersburg, Russia) for image analysis. The surface of the polished sections was etched to identify the microstructure of the metal in the following solution: 16 mL HNO_3_/16 mL HF/68 mL glycerol.

The polished sections of the samples were subjected to X-ray phase analysis (XRD) on a Rigaku Ultima IV diffractometer (Rigaku, Tokyo, Japan). The used radiation was Cu K_α_. The chemical composition of the samples’ structural components was analyzed on a JSM-6460LV scanning electron microscope (JEOL, Tokyo, Japan) equipped with an energy dispersive spectrometer (Oxford Instruments, Abingdon, United Kingdom) for qualitative and quantitative X-ray microanalysis (XRMA).Microhardness of the polished sections was measured on FM–800 microhardness tester (Future-Tech Corp, Kawasaki, Japan) at the load of 300 g.

## 3. Results

### 3.1. Structure and Microstructure

Figure 2 shows details No. 1 and 2, which were manufactured without thermal insulation of the process chamber. Numerous macrocracks are visible on the parts; these “defective” products were not further studied.

Cracking was observed immediately upon the end of the manufacturing process when the parts were cooled down. Based on this, we assumed that cracking is caused by internal thermal stresses preconditioned by the accelerated cooling of the detail. In order to reduce thermal stresses when the detail was cooled down, we made a process engineering decision to heat insulate the external part of the process chamber.

Details No. 3, 4 and 5 manufactured after the complex was updated had no visible cracks, pores and other defects (Figure 3). The wall thickness for all the parts is 4 mm. The cut-up sketches (see Figure 3) show the studied surfaces, on which polished sections were made.

During the microstructural metallographic examination of the polished sections of the samples of detail No. 3, no metal discontinuity was found. The polished sections of the samples of detail No. 4 have single pores of up to 40 μm (Figure 4a). During the microstructural metallographic examination of a polished section of the substrate, shrinkage pores of 50 to 200 μm are found on the inner side on the sample of detail No. 5 (Figure 4b), no defects are found on the polished section of the deposited part.

It should be noted that the structure of the samples of details No. 3 and 4 lacks visible boundaries of transitions from one deposited layer to another one, while the sample of detail No. 5 has such boundaries. After etching on the detail No. 5, layers of the deposited metal are clearly visible, while on other details this is not observed (Figure 5).

All the studied samples have a microstructure typical of the hardened state of two-phase titanium alloys: α-phase precipitates along the primary β-phase grain boundaries, needle-like α -phase precipitates in the grain body with thin β-phase interlayers between them (Figure 6). The so-called “basket weave” structure is clearly distinguished in some fields of view.

### 3.2. XRD Analysis

The XRD results (Figure 7) confirm the presence of two phases (α and β) in the structure, but there are differences in the quantitative ratio of these phases in the studied samples. According to the calculations, sample No. 3–2 (detail No. 3) contains 93% of α-phase and about 7% of β-phase; sample No. 4–2 (detail No. 4) contains 89% of α-phase and about 11% of β-phase; and 73% of α-phase and about 27% of β-phase were formed in the deposited part of sample No. 5.

### 3.3. XRMA Analysis

The XRMA method (Table 2) was used to analyze the chemical composition of various sections of the details manufactured by the laser melting method, the homogeneity of the chemical composition throughout the height of the product, as well as the composition of individual structural components. The general element-by-element analysis of the samples throughout the height of the detail was carried out by scanning the sections at an increase of 100 times from at least 5 sections for each of the samples (Table 2 shows the averaged data). The chemical composition of the structural components was also analyzed at different sections throughout the height of the product wall at an increase of 500 times (Table 2 shows the averaged data). According to the data of Table 2, β-phase is enriched with iron and vanadium.

### 3.4. Microhardness HV

The results of measuring the microhardness *HV* of the manufactured products are shown in Figure 8. According to Figure 8 the microhardness for detail No. 3 is change from 393 *HV* in the lower part to 413 *HV* in the upper part along the wall height; for detail No. 4—from 400 to 500 *HV*; for the deposited part of detail No. 5—from 420 to 505 *HV*. Moreover, it should be noted that for detail No. 3, a marked increase in hardness is observed at the last 15 mm of the wall height; on detail No. 4 stepped growth of microhardness is observed with jumps of 25 mm and 45 mm in height of the wall; microhardness along the height of the wall of detail No. 5 increases uniformly.

The distribution of microhardness in the place of melting the deposited metal and the substrate of detail No. 5 is shown in Figure 9. Microhardness values increase from 365 (for the substrate) to 440 (for the deposited part).

## 4. Discussion

The analysis of the microstructure throughout the height of the details walls shows that in the upper part of the products the structure of the α-phase lamellar precipitates is more dispersed than in the lower sections (see Figure 6). A similar regularity was noted in the works dealing with various methods of creating titanium alloy parts: SMD, DMD, LMD [22,25,26].

An explanation of this phenomenon of structure formation was discussed in works [28,29]. Such a gradient of the structure was explained by repeated heating when the laser passes over a given point, since the detail is formed layer by layer. Thus, the lower layers experience multiple heating cycles when applying the upper layers.

So, the microstructure of the details, which were obtained in the course of this study, is not uniform throughout the height of the wall, which is explained by the production technology (the lower layers of details are exposed to larger thermal effects than the upper ones).

When analyzing the microstructure, it was necessary to take into account the diagram of phase transformations during the continuous cooling of the β-solid solution from a temperature of 1050 °C. In our case, a martensitic transformation took place, which indicates the cooling rates were above 410 K/s [30].

This work was not targeted on fixing the dependences of the temperature distribution in the wall of detail when the layers are deposited, but, nevertheless, the obtained results on the structure dispersity are coherent with the hypotheses put forward by the aforementioned authors.

The quantitative ratio of the structural components by XRD analysis can be explained both by the composition of the samples (see Table 1) and by the peculiar features of phase transformations when the parts are cooled down.

It can be seen that the material’s hardness (see Figure 8) has slight fluctuations in the height of the product wall, however, in the upper part, the microhardness indicators are higher, which can be correlated with the structure of the samples, similar conclusions came and the authors of [22,23,24]. The difference in microhardness between samples 3 and 4 is explained by the geometry of the products. Detail No. 4 is two times higher than detail No. 3. Accordingly, the wall of detail No. 4 was manufactured in a much longer time and it was subjected to a longer thermal effect when the layers were built up.

When analyzing the results of microhardness measurements along the fusion line of the weld part and the substrate for detail No. 5 (see Figure 9), a gradient of values is observed: the *HV* values for the substrate are 20% lower than for the weld part. Such a gradient of hardness should be explained more likely not by the technology of manufacturing the substrate, but by the type of material from which it is made. The substrate of detail No. 5 was made of Ti–6Al–4V alloy powder, and the part itself was made of Ti–6Al–4Mo–1V powder (see Table 1).

According to the results of measuring the hardness of the height of the walls of details No. 3, 4 and 5 (see Figure 8), the hardness of details made of Ti–6Al–4V alloy is about 400–450 *HV*, and detail made of Ti–6Al–4Mo–1V alloy—450–500 *HV*, which corresponds to the difference in hardness of 10–20%. The observed lower values of the hardness of the substrate material of detail No. 5 (365 *HV*) can be explained by the fact that the Ti–6Al–4V alloy has higher values of the phase transformation temperature α+β↔β (950–1000 °C), than alloy Ti–6Al–4Mo–1V (920–960 °C). Consequently, the area of the substrate adjacent to the weld part will not experience multiple phase recrystallizations (unlike the material of the part wall) and will have a less dispersed, softer structure.

## 5. Conclusions

The experiments carried out according to the developed procedure demonstrate that a slight upgrading (installing an insulated process chamber) of the standard FL-Clad-R-4 laser melting complex can result in the fact that parts of a complex geometrical shape can be manufactured from titanium alloys using the combined SLM and DMD methods.

All the studied samples have a microstructure typical of the hardened state of two-phase titanium alloys: α-phase precipitates along the primary β-phase grain boundaries, needle-like α-phase precipitates in the grain body with thin β-phase interlayers between them. The microstructure of the details is not uniform throughout the height of the wall, which is explained by the production technology (the lower layers of details are exposed to larger thermal effects than the upper ones). This affects the microhardness index for the parts obtained: for the upper part of the wall, the microhardness indicators are slightly higher: the microhardness in the upper part of the parts is 10–25% higher (about 500 *HV*) than in the lower part (about 400 *HV*).

According to the study, the most vulnerable place of the products manufactured by combining the two methods is the melt junction of the substrate and the part. Since we observed here shrinkage defects in the form of pores, we can recommend to begin melting on a substrate at a low motion speed of the laser head. This is relevant, first of all, for the manufacture of details of an arbitrary geometrical shape, when the substrate is an integral structural part of the product.

The comparability of the obtained results with other known procedures of additive technologies in manufacturing titanium alloy products demonstrates the undoubted competitiveness of the method presented in this paper.

## Figures and Tables

**Figure 1 materials-12-03269-f001:**
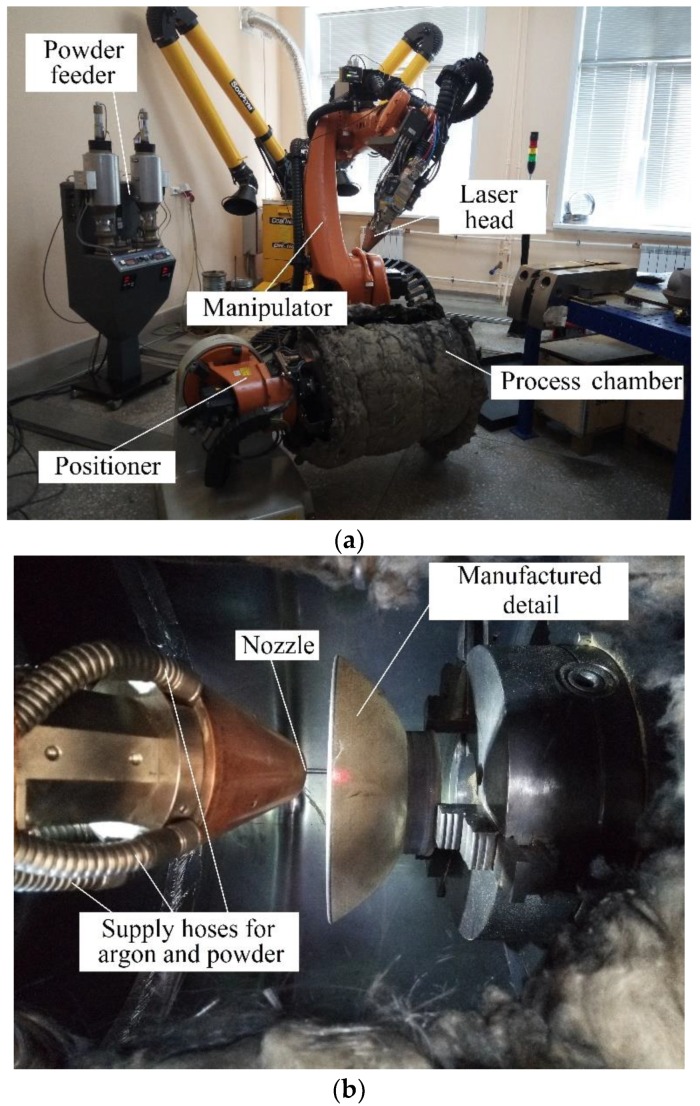
FL-Clad-R-4 complex of laser melting of metals (for DMD method): (**a**) General appearance; (**b**) Inner space of the process chamber.

**Figure 2 materials-12-03269-f002:**
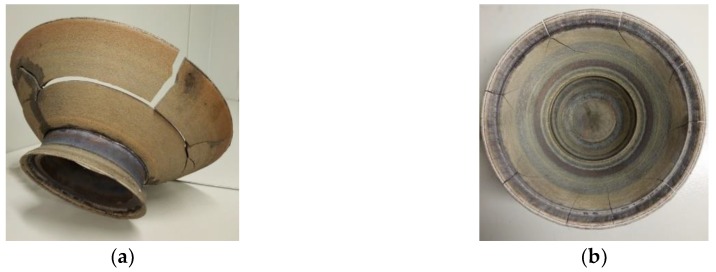
The appearance of the “defective” details: (**a**) No. 1; (**b**) No. 2.

**Figure 3 materials-12-03269-f003:**
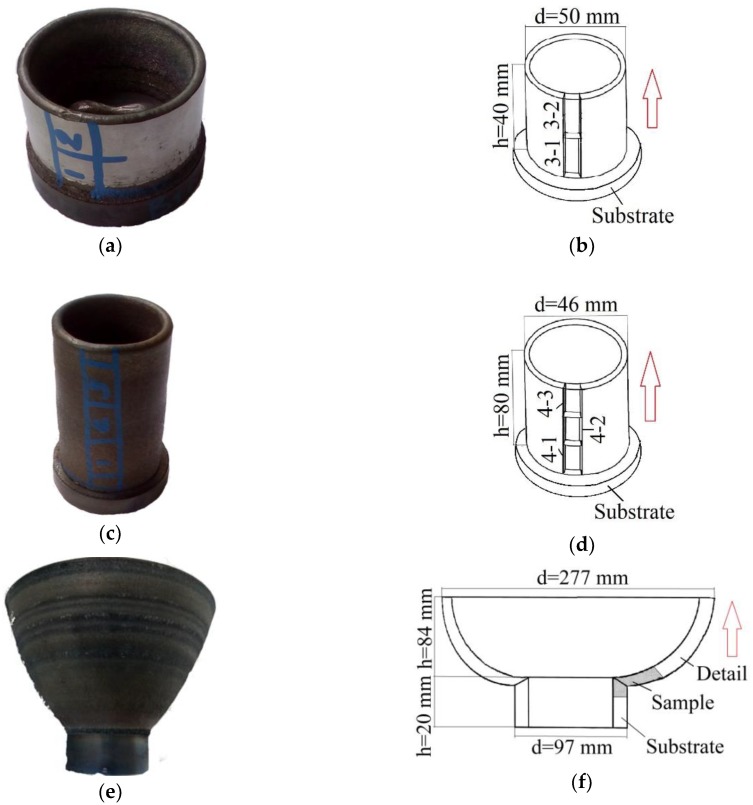
The appearance of the products and the cut-up sketch: (**a**,**b**) From detail No. 3; (**c**,**d**) From detail No. 4; (**e**,**f**) From detail No. 5. The arrows indicate the direction of application of the layers of manufactured details.

**Figure 4 materials-12-03269-f004:**
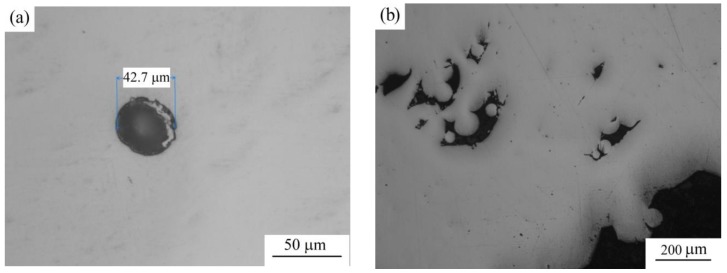
Pores on the thin sections: (**a**) For detail No. 4; (**b**) For area of joining of the substrate and the weld part on the sample of detail No. 5.

**Figure 5 materials-12-03269-f005:**
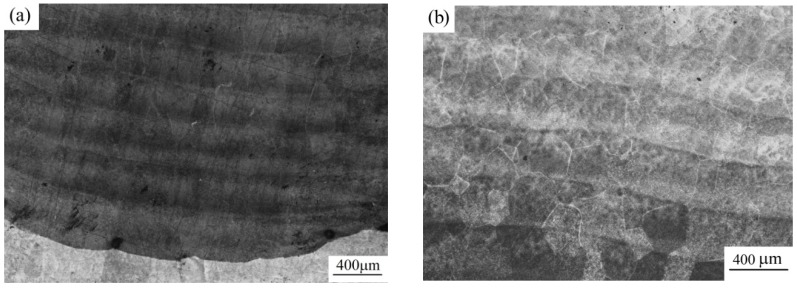
Layers of weld metal on the detail No. 5: (**a**) On the border with the substrate (the light area at the bottom of the image is the substrate); (**b**) In the main part of the paraboloid.

**Figure 6 materials-12-03269-f006:**
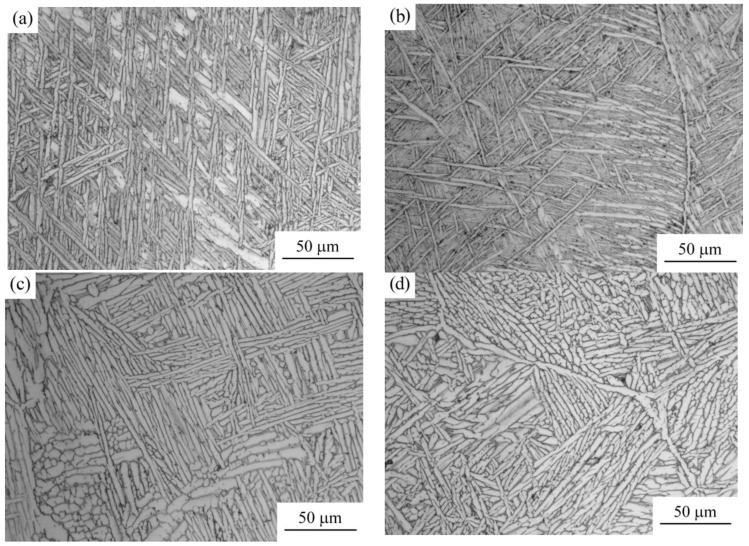
The microstructure of polished sections of the samples after etching according to optical microscopy (×500): (**a**) Sample No. 3–1; (**b**) Sample No. 3–2; (**c**) Sample No. 4–1; (**d**) Sample No. 4–2; (**e**) Sample No. 4–3; (**f**) Sample No. 5 (deposited part).

**Figure 7 materials-12-03269-f007:**
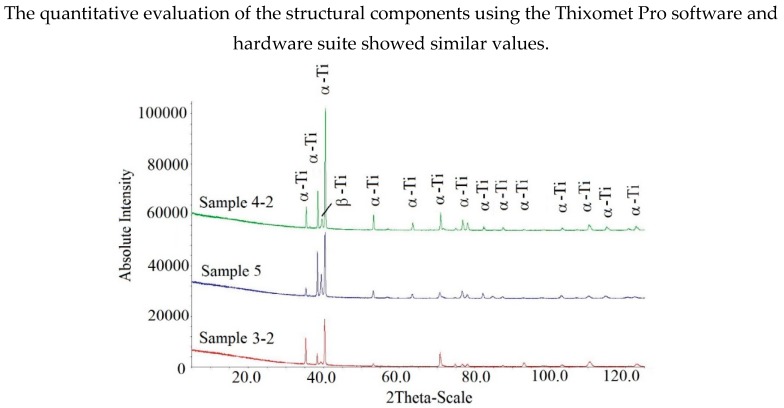
X-ray patterns of details No. 3 (sample No. 3–2); No. 4 (sample No. 4–2) and No. 5 (deposited section of the sample).

**Figure 8 materials-12-03269-f008:**
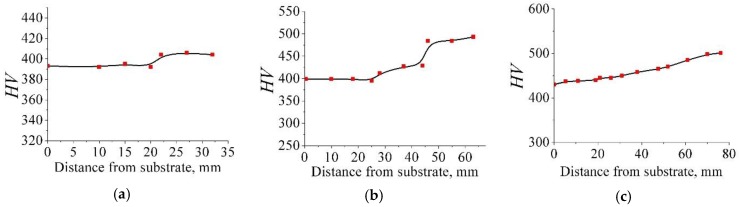
Microhardness measurement results: (**a**) Detail No. 3; (**b**) Detail No. 4; (**c**) Detail No. 5 (deposited part).

**Figure 9 materials-12-03269-f009:**
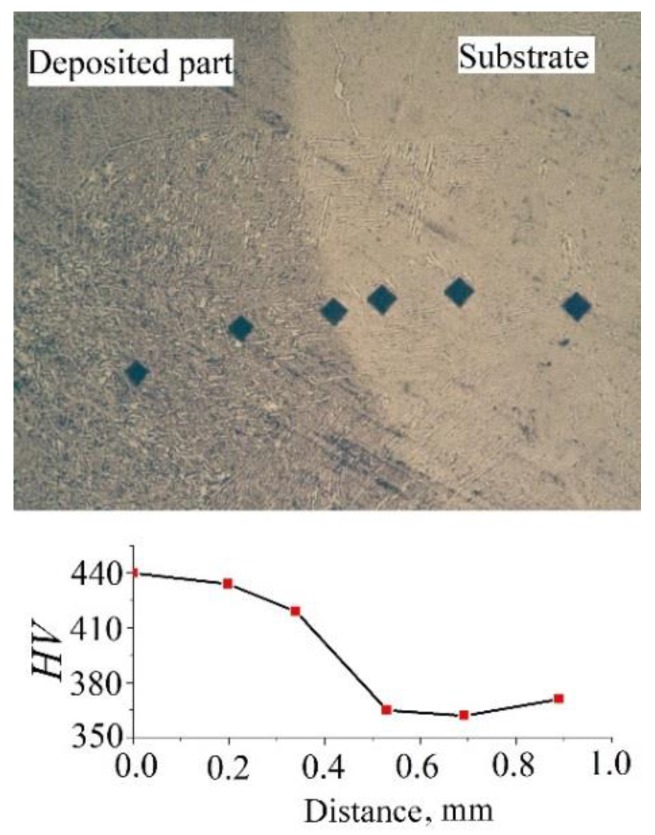
The distribution of hardness at the place of melting of the substrate and manufactured detail No. 5.

**Table 1 materials-12-03269-t001:** Compositions of metal powders, laser power (LP), powder feed speed (Vp), laser head displacement speed (Vl).

No.	Alloy Grade	Manufacturing Modes of Details				
		Method		LP, W	Vp, g/min	Vl, mm/sec
1, 2	Ti–6Al–4Mo–1V paraboloid	DMD	Without a heat insulator	1600	27	10
3, 4	Ti–6Al–4V cylinder	DMD	With a heat insulator			
5	Ti–6Al–4Mo–1V paraboloid	DMD				

**Table 2 materials-12-03269-t002:** Results of the XRMA, wt%.

Structural Components	Al	Ti	V	Fe	Zr	Mo
Detail No. 3						
Sample No. 3-2 (general analysis)	7.02	87.98	4.53	0.47	–	–
Sample No. 3-1 (general analysis)	6.57	89.52	3.91	–	–	–
α-phase (sample No. 3-2)	6.98	89.05	3.97	–	–	–
α-phase (sample No. 3-1)	6.69	89.76	3.55	–	–	–
β-phase (sample No. 3-2)	6.36	88.33	4.85	0.46	–	–
β-phase (sample No. 3-1)	6.43	88.26	4.53	0.78	–	–
Detail No. 4						
Sample No. 4-3 (general analysis)	6.61	88.91	4.48	–	–	–
Sample No. 4-1 (general analysis)	6.94	89.33	3.73	–	–	–
α-phase (sample No. 4-3)	6.71	91.25	2.04	–	–	–
α-phase (sample No. 4-1)	6.95	91.66	1.39	–	–	–
β-phase (sample No. 4-3)	5.14	84.07	9.54	1.26	–	–
β-phase (sample No. 4-1)	5.40	81.54	11.33	1.73	–	–
Detail No. 5						
Substrate	6.06	90.23	3.40	0.31	–	–
Bottom of the detail (at the substrate)	6.34	91.09	0.75	–	0.19	1.63
Top of the detail	6.57	90.24	1.19	–	–	2.00
α-phase (top part)	6.49	92.27	–	–	–	1.24
α-phase (bottom part)	6.94	91.25	1.04	0.24	–	0.53
β-phase (top part)	6.40	88.89	1.21	0.27	–	3.22
β-phase (bottom part)	6.49	88.36	2.66	0.68	–	1.81

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
