# Peer review of "A Study of the Structural Characteristics of Titanium Alloy Products Manufactured Using Additive Technologies by Combining the Selective Laser Melting and Direct Metal Deposition Methods"

_materials, 2019, doi:10.3390/ma12193269_

Round 1

Reviewer 1 Report

In this experimental paper, three different geometric samples are built and analysed (2 other samples are discarded). All samples are made with SLM and DMD so the experiment as designed is not suitable for evaluating the effectiveness of using SLM with DMD to improve quality (no control without SLM). The samples have different geometries and sizes, necessarily producing different thermal histories, build times etc.

The paper contains detailed analysis of the microstructure: micrographs, XRD, microhardness, XRMA at different heights in the samples. This is done properly and yields enough information for discussion.

There have previously been some excellent detailed analyses of Ti alloy microstructures during multilayer deposition and I found the discussion in this paper more a literature review of these papers. In the absence of composition differences, the microstructural and property differences between parts and within parts must have been due to differences in the thermal cycle but these are not considered and linked to microstructure. The authors should focus the Discussion mainly on their own work.

The Conclusion is not supported by the results or discussion. Micrographs do not show the microstructures indicated or evidence of pores at the melt junction. All samples were prepared with SLM so there is no like for like comparison to give evidence it is effective at controlling shrinkage defects.

Other points

In the Introduction (L46-58) sentences phrased in the form ‘In the work [25] studied…’ or ‘Authors of work [26] showed’ are not correct.  Instead, use (eg.) ‘In the work Dinda et al [25] studied…’ or ‘Kelly and Kampe [26] showed…’. Also L193-194, sentence beginning ‘To explain such a gradient of hardness…’ needs rewriting to correct English.

Figure 1: It is possible to determine the set up, but a schematic diagram and picture could be clearer than two pictures.

More information on the building of sample 5 would help better define the build process: does the nozzle orientate to the build direction of the wall or remain parallel to the axis of the paraboloid during the build? Is the 0.3 mm growth height of the wall in its build direction or parallel to the axis?

Author Response

Dear Reviewer, first of all, many thanks for the careful attitude to our manuscript.

Thank you for your valuable suggestions and comments.

Let us answer your сomments and suggestions.

Substrates of a simple geometric shape were made by the SLM method at the Composite JSC company (Korolev, Russia). However, the small size of the working camera of the equipment available at JSC Composite company did not allow to grow on these substrates the details of the necessary geometric shape specified by the technical specifications of the sizes. Therefore, further work was continued on the basis of South Ural State University (Chelyabinsk, Russia) in order to determine the possibility of combining SLM and DMD methods for the manufacture of parts from titanium alloys (including various compositions) of complex geometric shapes.

The thermal cycle in the production of the part has a significant effect on the microstructure. This is what we emphasize in our work.

We added pore photos to the article text (Fig. 4).

We made the changes you indicated in the text.

In the process of preparing the manuscript, we tried your preferred version of the presentation of Fig. 1, but it turned out to be difficult for perception. Therefore, we decided to show the equipment more clearly, using photographs.

According to your recommendation, we have added the assembly direction on the details diagrams (red arrows in Fig. 3).

Respectfully,

Team of Authors.

P.S. The editors agreed to assist us in editing the English translation of the manuscript.

Reviewer 2 Report

The language needs improvement to convey your ideas.

The methodology and results is not clear enough to justify your claims

The process parameters for SLM and DMD is confusing.

Author Response

Dear Reviewer, first of all, many thanks for the careful attitude to our manuscript.

Thank you for your valuable suggestions and comments.

Let us answer your сomments and suggestions.

Unfortunately, not only you, but also other reviewers pointed out inaccuracies in the description of the methodology. We made changes to the text of the article in order to make our methods and approaches more understandable, informative, and reproducible.

Respectfully,

Team of Authors.

P.S. The editors agreed to assist us in editing the English translation of the manuscript.

Reviewer 3 Report

In this paper, the authors fabricated several titanium objects (2 cylinders with different shapes, and a paraboloid) using additive manufacturing technology. The microstructure in different locations of the parts, phase composition, chemical composition, and hardness were studied. The information discussed in the paper is in the scope of the journal. However, the paper gives limited new scientific information toward additive manufacturing of titanium alloy when compared with previous research. The writing and presentation of some content in the paper need to be highly improved. The reviewer suggested major revision considering the following issues.

(1) The abstract section needs to be revised. It mostly shows how the work was performed but there is no brief research result presented. It hardly gives readers the unique information contributed by the paper.

(2) No first-person point narrative, such as “we”, “I”

(3) “Thus, in the literature there are still lacks a universal technology for manufacturing titanium alloys produced with the specified properties”. The “universal technology” here is not clear. It seems the mentioned studies provide approaches for making titanium parts. Besides, did the paper suggest a universal technology for fabricating titanium?

(4) Please list the chemical composition of the two materials used in the research.

(5) The title and contents of the paper show the study used combined SLM and DMD methods to fabricate titanium components. As the reviewer observed, the deposition system FL-Clad-R-4 complex shown in Figure 1 is more like a DMD system, where a laser was used to create a molten pool and powders are delivered into the molten pool through powder feeding nozzles. SLM is a powder-bed AM process in which layers of powders are sprayed on a big building plate and the powders are selectively melted and solidified to form the part. Since the title says “combined SLM and DMD methods”, both processes much play important roles in building the components. Could you show where and how SLM process is carried out using the FL-Clad-R-4 system? The table 1 shows the substrates are SLM fabricated. Are they built using the FL-Clad-R-4 system? If yes, please explain how to realize it using the deposition system. If not, please add information about the SLM system and process parameters.

(6) The substrates in table 1 are fabricated using SLM method. What are the benefits to use SLM to build the substrate, compared with building the entire part using DMD? The reviewer didn’t see the advantages using SLM as no such information was discussed in the paper. Such information must be added otherwise, there is no need to combine SLM and DMD.

(7) What is the difference between detail No. 1 and No.2? I see they are sharing the same material, same methods and same parameters. Are they just two parts from two runs?

(8) “It should be noted that the structure of the samples …… while the sample of detail No. 5 has such boundaries”. Could you mark the boundaries in Figure 4f? Also, did you claim there are no boundaries in sample No. 3 and No.4 from Figure 4a-e? The scale may be too small to observe the layer boundaries as the layer thickness is about 300 microns. The images might be taken in a single layer. Please double check.

(9) What information could readers get from table 2?

(10) There is no presentation of microhardness information in section 3.4. Please include what you can see from the figures.

(11) In figure 7, the hardness was not in a line. Since the x-axis of the figure is distance, the plot might be not clear. Please consider measure the hardness again.

(12) In conclusion, “the microhardness indicators are slightly higher”. The hardness values or range should be added.

(13) “The comparability of the obtained results……demonstrates the undoubted competitiveness of the method presented in this paper”. What are the competitiveness of the method compared with others’ work?

Author Response

Dear Reviewer, first of all, many thanks for the careful attitude to our manuscript.

Thank you for your valuable suggestions and comments.

Let us answer your сomments and suggestions.

We have made changes to the "abstract" section in accordance with your comments.

According to your second remark, we can say the following. The scientific school that we represent traditionally avoids such formulations. Since, in our opinion, this cannot significantly affect the quality of the material and its perception, we will retain our wording in tribute to our scientific mentors.

We absolutely agree with your third comment and have made changes to the wording.

In accordance with your fourth observation, we have added information on the chemical composition of the test material.

Thanks for the comment. We agree that this moment in the first version of the article was not spelled out exactly. Therefore, we have made appropriate changes to the text of the article.

Substrates of a simple geometric shape were made by the SLM method at the Composite JSC company (Korolev, Russia). However, the small size of the working camera of the equipment available at JSC Composite company did not allow to grow on these substrates the details of the necessary geometric shape specified by the technical specifications of the sizes. Therefore, further work was continued on the basis of South Ural State University (Chelyabinsk, Russia) in order to determine the possibility of combining SLM and DMD methods for the manufacture of parts from titanium alloys (including various compositions) of complex geometric shapes.

About the differences between the details #1 and #2. Indeed, these two details were made using the same technology and the same materials. And we purposefully presented both photographs with the aim to, firstly, show the very presence of defects and their different appearance, and secondly, to show that the appearance of cracks was not random, because parts were made in one mode, but in several attempts.

We added photos with layers of weld metal in the text of the article.

In the text of the article we provide a description and clarification of the information presented in table 2. We add a description of the graphs of hardness changes in the appropriate section of the article.

The software of the microhardness tester allows fixing the coordinates of each point at which the hardness was measured. Therefore, we present the actual projection of the measured points on the "X" axis.

We have added a range of hardness values to the indicated section.

To your question about the competitiveness of our method, we want to note the following. The proposed method allows without significant financial costs to produce parts of complex geometric shape (in particular a paraboloid) with the necessary structural and mechanical characteristics. Thus, the manufactured parts are competitive with parts manufactured by classical casting methods or through other additive technology techniques.

Respectfully,

Team of Authors.

P.S. The editors agreed to assist us in editing the English translation of the manuscript.

Reviewer 4 Report

The manuscript entitled “A study of the structural characteristics of titanium alloy products manufactured using additive technologies by сombined SLM and DMD methods” reports an effort to present a modified additive manufacturing (AM) method based on the laser cladding complex FL-Clad-R-4. To modify the FL-Clad-R-4 complex, the authors installed an insulated process chamber. To demonstrate the performance capabilities of the technique, the authors manufactured five thin-walled parts using two titanium alloys and studied their microhardness and microstructures using optical microscopy, XRD, and XRMA. The subject matter would be of potential interest to readers of Materials involved in metal AM research.

However, in my opinion, the research challenge is unclear and should be described more in detail. The authors state that a “universal methodology for manufacturing titanium alloys details using AM” is lacking and thus they combine selective laser melting (SLM) and direct metal deposition (DMD) for developing such a methodology (p. 2, lines 61-65). It might help the reader if you explained what exactly was taken from SLM technology and how the insulated process chamber helps by manufacturing better parts of titanium alloys, as, e.g., there is a growing body of literature on wire + arc additively manufactured Ti-6Al-4V (Cranfield University, UK) possessing reliable mechanical properties and produced without a specific environment.

Below I give some comments and suggestions:

This paper needs a revision by a native English proofreader. In the current state, the manuscript is confusing for the reader. Page 2, Table 1: Please clarify what is the reason for choosing different titanium alloys for a paraboloid and a cylinder. I would also suggest making the table more compact as there is no difference between 1, 2, and 5 specimens as well as between 3 and 4 ones, except for their numbers. Page 5, Fig. 4: DED Ti64 parts are known to suffer from large columnar grains (prior β-structure). It would be interesting to consider this macrostructure in your study as well. Apart from that, I would recommend adding the build direction in the figure. Page 8, lines 155-157: It might help the reader if you added a scheme explaining where the samples of the “inner side” and “deposited part” were taken. Page 8, lines 188-191: It would be interesting to discuss the reasons for the difference in microhardness between 3 and 4 specimens (the maximum value for the 3 specimen is around 400, while that for the 4 specimen is around 500).

Author Response

Dear Reviewer, first of all, many thanks for the careful attitude to our manuscript.

Thank you for your valuable suggestions and comments.

Let us answer your сomments and suggestions.

We have made a number of refinements to the text of the article concerning:

Substrates of a simple geometric shape were made by the SLM method at the Composite JSC company (Korolev, Russia). However, the small size of the working camera of the equipment available at JSC Composite company did not allow to grow on these substrates the details of the necessary geometric shape specified by the technical specifications of the sizes. Therefore, further work was continued on the basis of South Ural State University (Chelyabinsk, Russia) in order to determine the possibility of combining SLM and DMD methods for the manufacture of parts from titanium alloys (including various compositions) of complex geometric shapes.

An insulated process chamber was proposed as the optimal solution for the manufacture of quality details. Because the manufacture of details of an arbitrary geometric shape implies uneven heat transfer from different parts of the product. What leads to cracking (see Fig. 2) when cooled in air without a “specific environment”.

The choice of alloy grade was determined by the terms of reference for the manufactured parts and the requirements for their properties.

Thanks for the recommendations on how to present Table 1. They helped us a lot.

We have added to the article text illustrations of deposited layers.

The pattern of cutting samples for research and the direction of application of the layers are presented in Fig. 3.

The difference in microhardness between samples 3 and 4 is explained by the geometry of the products. Part 4 is two times higher than part 3. Accordingly, the wall of part 4 was manufactured in a much longer time and it was subjected to a longer thermal effect during layers build up. We added relevant comments in to the text of the article.

Respectfully,

Team of Authors.

P.S. The editors agreed to assist us in editing the English translation of the manuscript.

Round 2

Reviewer 1 Report

Authors better explain what was done : SLM followed by DMD. Based on this I would suggest renaming the paper slightly to ‘….Technologies by Combining SLM and DMD Methods’ or ‘….Technologies by SLM and DMD Methods’

The discussion in this paper is still too much about others papers. Sentences begin ‘In [29]…’, ‘Thus [28] presents…’, ‘According to the data of [29]…’ etc    The authors should focus the Discussion mainly on their own work. For example, instead of writing: ‘ According to the results of [30], when the cooling rates are above 410 K/s, martensitic transformation is realized; and when the rates are less than 410 K/s, it results in diffusion transformations leading to the formation of the Widmannstät structure. In our case, the transformation of the martensitic type is realized.’ It would be better to write ‘In our case, a martensitic transformation took place, which indicates the cooling rates were above 410 K/s [30].’ (This focusses on the work done and explaining what happened in these experiments).

The conclusions correctly states ‘The microstructure of the details is not uniform throughout the height of the wall, which is explained by the production technology (the lower layers of details are exposed to larger thermal effects than the upper ones)’ but this needs to be shown in the Discussion.

Author Response

Good Day, Dear Reviewer!

Thanks for the additional comments.

Below we provide answers to them.

We changed the title of the manuscript. We significantly reduced the discussion section, removing from it a detailed description of the results of other authors and focusing on our work. We added the phrase about the heterogeneity of the structure along the height of the wall in the discussion section.

Respectfully,

Team of Authors.

Reviewer 3 Report

It can be accepted for publication.

Author Response

Good Day, Dear Reviewer!

Thank you for recommendation to the publication of our manuscript.

Respectfully,

Team of Authors.